# Expression of ERBB Family Members as Predictive Markers of Prostate Cancer Progression and Mortality

**DOI:** 10.3390/cancers13071688

**Published:** 2021-04-02

**Authors:** Sylvie Clairefond, Véronique Ouellet, Benjamin Péant, Véronique Barrès, Pierre I. Karakiewicz, Anne-Marie Mes-Masson, Fred Saad

**Affiliations:** 1Centre de Recherche du Centre Hospitalier de l’Université de Montréal (CRCHUM) et Institut du Cancer de Montréal (ICM), Montreal, QC H2X 0A9, Canada; sylvie.clairefond@umontreal.ca (S.C.); veronique.ouellet.chum@ssss.gouv.qc.ca (V.O.); benjamin.peant.chum@ssss.gouv.qc.ca (B.P.); veronique.barres.chum@ssss.gouv.qc.ca (V.B.); pierre.karakiewicz@umontreal.ca (P.I.K.); anne-marie.mes-masson@umontreal.ca (A.-M.M.-M.); 2Département de Médecine, Faculté de Médecine, Université de Montréal, Montreal, QC H3T 1J4, Canada; 3Département de Chirurgie, Faculté de Médecine, Université de Montréal, Montreal, QC H3C 3J7, Canada

**Keywords:** prostate cancer, predictive biomarkers, biochemical recurrence, development of bone metastases, cancer-specific mortality, immunofluorescence, tumor glands

## Abstract

**Simple Summary:**

Patients diagnosed with prostate cancer are usually offered a standard treatment plan based on their Gleason score, stage, and prostate-specific antigen (PSA) level. However, studies on other cancers have shown the importance of using biomarkers in addition to clinical and pathologic parameters to personalize therapeutic decisions. Given the important heterogeneity in the natural history of localized prostate cancer, novel prognostic biomarkers would aid in patient stratification and decision making. Here, our study shows that members of the ERBB family are markers that have high prognostic value for predicting biochemical relapse, metastasis development, and even prostate cancer-related mortality. The integration of these markers into clinical practice may eventually lead to more appropriate therapeutic decisions in newly diagnosed patients and potentially reduce prostate cancer morbidity and mortality.

**Abstract:**

Background: EGFR, ERBB2, ERBB3, and ERBB4 are growth receptors of the ERBB family implicated in the development of epithelial cancers. Studies have suggested a role for EGFR and ERBB3 in the development of prostate cancer (PC), while the involvement of ERBB2 and ERBB4 remains unclear. In this study, we evaluated the expression of all members of the ERBB family in PC tissue from a large cohort and determined their contribution, alone or in combination, as prognostic markers. Methods: Using immunofluorescence coupled with digital image analyses, we quantified the expression of EGFR, ERBB2, ERBB3, and ERBB4 on radical prostatectomy specimens (*n* = 285) arrayed on six tissue microarrays. By combining EGFR, ERBB2, and ERBB3 protein expression in a decision tree model, we identified an association with biochemical recurrence (log rank = 25.295, *p* < 0.001), development of bone metastases (log rank = 23.228, *p* < 0.001), and cancer-specific mortality (log rank = 24.586, *p* < 0.001). Conclusions: Our study revealed that specific protein expression patterns of ERBB family members are associated with an increased risk of PC progression and mortality.

## 1. Introduction

Prostate cancer (PC) is one of the most commonly diagnosed and lethal cancers in men worldwide. PC is a heterogeneous disease encompassing low- (slow and non-aggressive progression) and high-risk (rapid progression) diseases. Approximately a quarter of patients will develop the latter, characterized by the development of metastasis and subsequent death [1]. Currently, clinicians use prostate-specific antigen (PSA) levels, Gleason score, and clinical stage in an attempt to predict patient prognosis and guide treatment options [2,3]. However, these clinical parameters are insufficient to reliably distinguish between low- and high-risk diseases. In this context, the discovery of prognostic biomarkers to implement new indicators is needed [4].

In our laboratory, we identified several biomarkers associated with an increased risk of biochemical recurrence (BCR), such as p65 nuclear frequency [5,6], PTEN [7], CD73 [8], and PUMA-NOXA [9]. Additional biomarkers were also identified with an increased risk of development of bone metastases, e.g., nuclear p65 expression [10] and CCN3 [11], and the ability to predict PC specific mortality, such as nuclear p65 expression [10] and neutral endopeptidase CD10 [12]. Research related to biomarkers is growing in number and several seem very promising; however, to date, no tissue biomarker is routinely used in the clinical setting.

The epidermal growth factor family of protein include EGFR, ERBB2, ERBB3, and ERBB4. Their structure is composed of an extracellular binding domain, a transmembrane domain, and an intracellular domain with tyrosine kinase activity [13]. The homology of these proteins vary between 40% and 50%, with the highest homologies located in the intracellular and the lower homologies associated with the extracellular domain, which provides specificity and affinity to their specific ligands [14]. These transmembrane glycoproteins play an important role in multiple cellular pathways, including migration, proliferation, metabolism, differentiation, and survival [15,16,17]. The mechanism of action of these receptors begins with the binding of a ligand to the extracellular domain, leading to a homo- or heterodimerization with another ERBB receptor, resulting in the subsequent auto- or trans-phosphorylation of the intracellular domain. The wide variety of downstream cellular responses is mainly due to the multiple dimerization possibilities and phosphorylation sites. Although there is no known ligand of ERBB2 and the kinase activity of ERBB3 is minimal, signal transduction of the heterodimer ERBB2–ERBB3 does occur [18].

Due to their important cellular functions, mutations or dysregulation of ERBB expression could have dramatic effects. Several studies have highlighted their involvement in the development and progression of several human cancers [15,19], including PC [20,21,22]. Indeed, high expression of EGFR is associated with poor PC patient prognosis [22,23,24]. Controversial results have been obtained for other members of the family [22,25,26,27,28,29]. Moreover, many studies have sought to target individual members of the ERBB family to develop anti-cancer therapies [15,30].

In this study, we quantified, for the first time, the protein expression of all four members of the ERBB family on a single tissue microarray-based cohort of radical prostatectomy specimens to determine their usefulness as prognostic markers in PC alone or in combination.

## 2. Materials and Methods

### 2.1. Cell Lines and Tissue Culture Conditions

Jurkat T lymphoma cells were kindly provided by Dr. Lapointe Réjean (CRCHUM), while MCF-7, SKOV3, and all PC cell lines (22Rv1, LNCaP, DU145, and PC3) were obtained from the American Type Culture Collection (ATCC, Manassas, VA, USA). Jurkat T lymphoma cell and PC cells were maintained in RPMI 1640 medium (Wisent Inc., St-Bruno, QC, Canada), MCF-7 was grown in DMEM medium (Wisent Inc.), and SKOV3 in OSE medium (Wisent Inc.). All culture media were supplemented with 10% fetal bovine serum (FBS) (Gibco^®^, Thermo Fisher Scientific, Waltham, MA, USA), 0.454 μg/mL of amphotericin B (Wisent Inc.), and 90 μg/mL gentamycin sulfate (Wisent Inc.).

### 2.2. Creation of Cell Line Pellets

Cell line pellets were used as a control on tissue microarray (TMA) and prepared as previously described [31]. This method was developed to fix and embed cell suspensions in paraffin using HistoGel™ (Thermo Fisher Scientific) to ensure high cell density per core when arrayed on a TMA. The embedded cell suspensions in paraffin made it possible to reproduce the same conditions with which the patient samples were tested.

### 2.3. Western Blot Analysis

Whole-cell protein extracts were prepared using lysis buffer (1% Triton, 10% glycerol, 50 mM Tris, 2 mM EDTA, and 150 mM NaCl) supplemented with fresh protease inhibitors (PIA32961, Thermo Fisher Scientific) and incubated 1 h at room temperature, followed by centrifugation. Proteins were dosed using a Bradford assay (Bio-Rad, Hercules, CA, USA). A total of 30 μg of whole-cell lysate was loaded on 6% sodium dodecyl-sulphate-polyacrylamide gel electrophoresis (SDS-PAGE, Bio-Rad) and transferred to nitrocellulose membrane using the Trans-Blot Turbo Transfer System (Bio-Rad). The membrane was immunoblotted with either rabbit monoclonal anti-EGFR (1:10,000, EP38Y, ab52894, Abcam Inc. Cambridge, UK), mouse monoclonal anti-ERBB2 (1:750, 3B5, OP15L, Calbiochem, San Diego, CA, USA), rabbit monoclonal anti-ERBB3 (1:200, D22C5, #12708, Cell Signaling Technology, Danvers, MA, USA), or mouse monoclonal anti-ERBB4 (1:500, C-7, sc8050, Santa Cruz Biotechnology, Dallas, TX, USA) antibodies. Each primary antibody was diluted in Tris-buffered saline tween 20 (TBS-T) containing 5% fat-free milk powder. α-tubulin was used as a loading control (DM1A, sc-32293, Santa Cruz Biotechnology). Immunoreactive bands were detected by enhanced chemiluminescence (ECL, GE Healthcare, Little Chalfont, UK).

### 2.4. Patient Cohort

The TF123 cohort included 300 primary PC patients who underwent radical prostatectomy at the Centre hospitalier de l’Université de Montréal (CHUM, Montréal, QC, Canada) between 1993 and 2006. Each patient signed an informed consent form for their participation in the Centre de recherche du Centre hospitalier de l’Université de Montréal (CRCHUM) PC biobank. The CRCHUM ethics review committee approved the study. A total of 15 patients were excluded due to preoperative hormone therapy. The time to BCR was defined as the time interval between the date of surgery and an increase in PSA levels above 0.2 ng/mL and rising, or when a decision to institute additional therapy was made.

### 2.5. Construction of TMA

A specialized genitourinary CHUM pathologist identified and traced out regions of cancer (Tumor: T), as well as adjacent non-cancerous areas (adjacent benign: BA) on fresh hematoxylin and eosin-stained slides obtained from formalin-fixed paraffin-embedded (FFPE) specimens. Two or three cores (0.6 mm) of BA and T were arrayed on two separate recipient blocks using a TMA array (Pathology Devices, Inc., Westminster, MD, USA). This TMA series (TF123) was composed of a total of six TMA blocks.

### 2.6. Immunofluorescence

For each biomarker, a TMA section of 4 µm was subjected to semi-automatic immunofluorescence (IF) multiplex staining protocol using the Benchmark XT auto-stainer (Ventana Medical Systems, Tucson, AZ, USA). These protocols include standard steps of deparaffinization, hydration/dehydration, and washes. The antigen retrieval was performed in Cell Conditioning #1 solution (#950-124, Ventana Medical Systems). Primary antibodies diluted in phosphate-buffered saline (PBS) (EGFR 1:50, ERBB2 1:650, or ERBB4 1:50) or in signal stain antibody diluent (#8112, Cell Signaling Technology, Danvers, MA, USA) (ERBB3 1:10) and were incubated at 37 °C for 60 min. Slides were blocked with a protein block serum-free solution (DAKO, Agilent, Santa Clara, CA, USA) during 20 min. Following this step, sections were removed from the auto-stainer away from light. Secondary fluorescent antibodies (1:250, Cy5™ goat anti-rabbit IgG or Cy5™ goat anti-mouse IgG from Thermo Fisher Scientific) diluted in PBS containing 1% bovine serum albumin (BSA, Sigma-Aldrich, St. Louis, MO, USA) were incubated at room temperature for 45 min. To avoid cross-reactivity, slides were blocked overnight with a mouse-on-mouse blocking reagent (MKB-2213, Vector Laboratories Inc., Burlingame, CA, USA) diluted 1:5 in PBS.

To detect the epithelium, a cocktail of antibodies against cytokeratins 8 (1:100, TS1, MA5-14428, Thermo Fisher Scientific) and 18 (1:100, DC-10, sc-6259, Santa Cruz Biothechnogy, Dallas, TX, USA) (used for EGFR and ERBB3) or a ready-to-use mix of cytokeratins 8 and 18 (1:2, Flex, clone EP17/30, DAKO, Agilent) (used for ERBB2 and ERBB4) were used. These antibodies were diluted in phosphate-buffered saline (PBS) and incubated at room temperature for 60 min. This step was followed by incubation with a secondary fluorescent antibody (1:250, Alexa Fluor^®^ 546 donkey anti-mouse IgG or Alexa Fluor® 546 donkey anti-rabbit IgG, Thermo Fisher Scientific).

To properly identify basal cells, a cocktail containing antibodies against p63 (1:650, 4A4, Ab-1, Neomarkers, Fremont, CA, USA) and high molecular weight cytokeratin (1:50, 34bE12, CLSG36689-05, Cedarlane, Fremont, CA, USA) was applied for 45 min to the section, and this was followed by the secondary fluorescent antibody Alexa Fluor^®^ 488 goat anti-mouse IgG (1:250, Thermo Fisher Scientific). Following a DAPI staining, to identify nuclei, each slide was incubated for 15 min at room temperature with a 0.1% solution of Sudan Black B (Research Organics, Cleveland, OH, USA) in 70% ethanol to quench tissue autofluorescence.

Finally, slides were mounted using Fluoromount™ Aqueous Mounting Medium (F4680, Millipore Sigma, Burlington, MA, USA). A negative control slide was performed in parallel (one for each biomarker) and incubated with PBS instead of the primary antibodies, then processed with the appropriate secondary antibodies.

### 2.7. Digital Image Analyses and Pre-Processing of Scoring Data

All slides were scanned within 24 h with a 20× Olympus Optical microscope BX61VSF (Olympus, Shinjuku, Tokyo, Japan) and visualized with OlyVIA software (Olympus). Scanned images were imported to VisiomorphDP software (Visiopharm, Hoersholm, Denmark). This software allows the development of semi-automated analysis protocol packages (APPs) to determine the expression levels of each biomarker by the mean fluorescence intensity (MFI) in each compartment (i.e., stroma and epithelium cytoplasm) [6].

We performed quality control of the tissue cores to exclude those that were damaged during the processing or cores containing less than 5% of epithelial cells. Duplicate cores presenting with significant differences were identified with scatter plots and Mann–Whitney test using GraphPad Prism software V6 (GraphPad, La Jolla, CA, USA). We then reviewed the images to determine if the difference observed was due to a technical issue. In such cases, the core was excluded from the analysis. However, data were kept if no unspecific staining anomaly was noted. To properly compare all TMAs together, the mean fluorescence intensity values of each core were normalized according to a calculated ratio. This ratio results from the mean fluorescence intensity across all TMA sections for a given biomarker divided by the mean fluorescence intensity (biomarker) for a given slide.

### 2.8. Statistical Analysis

Statistical analyses were performed with SPSS Statistics 25.0 software package (SPSS Inc., Chicago, IL, USA). To compare biomarker expression between tissue compartments (epithelial versus stroma), a Mann–Whitney test was used. To identify the appropriate threshold for survival analyses, data were displayed as quartiles to explore data trends and identified the percentile providing the best dichotomization for each biomarker. Survival analyses were performed using the Kaplan–Meier method coupled with a log-rank test. Univariate and multivariate Cox regression analyses were used to estimate the hazard ratios (HR) for each biomarker. A two-sided *p*-value <0.05 was considered statistically significant. The construction of the decision tree model was done using R software version 3.4.3 with RPART package (R Core Team, R Foundation for Statistical Computing, Vienna, Austria).

## 3. Results

### 3.1. Antibody Validation in PC Cell Lines

Although all antibodies have already been reported in the literature, we validated their specificity in a Western blot assay. We observed that all antibodies showed specific bands (Appendix A). We noted that the EGFR protein levels were higher in DU145, 22Rv1, and PC3 cells when compared to the LNCaP cell line. PC cell lines only weakly express ERBB2 with greater expression in LNCaP and 22Rv1 cells. No ERBB3 expression was detected in LNCaP or PC3 cells, while 22Rv1 and DU145 cells presented high expression. Finally, only the 22Rv1 cell line expressed ERBB4. Jurkat T lymphoma cells were used as control (negative) for ERBB receptors along with the well characterized MCF-7 (EGFR-, ERBB2-, ERBB3+, and ERBB4+) and SKOV3 (EGFR+, ERBB2+, ERBB3 weak, and ERBB4+) cell lines (Appendix A). Since in this study, we used these antibodies in formalin-fixed paraffin-embedded tissue, we created, fixed, and embedded cell pellets from these cell lines and performed an immunofluorescence (IF) assay. We noted that the expression of the ERBB family members was similar to those observed in the Western blot (Appendix A).

### 3.2. Patient Characteristics and Clinical Parameters

The analyzed TF123 TMA series was composed of 285 PC patients who did not receive neoadjuvant androgen deprivation therapy before radical prostatectomy. This was a mature cohort with a median follow-up of 129 months. Their demographic, histopathological, and clinical parameters are detailed in Table 1. The incidence of BCR at five years was 33% (94 patients), the incidence of bone metastasis at 10 years and death specific mortality were 6.3% (18 patients).

### 3.3. ERBB Family Member’s Expression in Human PC Specimens

To assess the usefulness of the ERBB family members as PC prognostic markers, we performed a multiplex IF assay incorporating one receptor (red: EGFR, ERBB2, ERBB3, or ERBB4) with specific masks to define the epithelium (yellow: CK8/18), the basal cells (green: p63/CKHMW, present in benign/normal prostate glands), and the nucleus (blue: DAPI) on the TF123 TMA series.

As expected, EGFR, ERBB2, ERBB3, and ERBB4 presented a membrane and cytoplasmic localization in the epithelium of both T (Figure 1A) and BA tissue cores. EGFR expression was significantly higher in BA compared to T tissue (*p* < 0.0001, MFI = 705 vs. 654, Figure 1B) as opposed to ERBB2 (*p* = 0.0230, MFI = 112 vs. 115, Figure 1C) and ERBB3 (*p* < 0.0001, MFI = 1126 vs. 1163, Figure 1D). Since we observed a low level of expression of ERBB4 in the PC cell lines and their derivates, and to avoid any waste of material, we decided to stain only one TMA slide with ERBB4. Despite the specificity of the antibody used, ERBB4 did not show a clear signal compare to background staining and had an MFI similar to the negative control (Figure 1E). Therefore, we did not include the analyses with ERBB4. We also noted that the expression of all receptors was significantly weaker in the stroma when compared to the epithelium (Figure 1B–E).

### 3.4. EGFR, ERBB2, and ERBB3 Expression Is Associated with an Increased Risk of BCR at 5 Years

To quantitate the expression of each biomarker, we performed digital image analysis of each core using an algorithm that detected only the epithelial compartment. This algorithm targets the region of epithelial cells stained by the cytokeratins (CKs) cocktail used for the detection of epithelial cells then measures the fluorescence intensity in the channel corresponding to the marker of interest.

To assess the prognostic capacity of the ERBB, we first evaluated if they were associated with BCR (less than five years). To determine the appropriate threshold for each biomarker to dichotomize their expression levels, we used the quartiles methods. Using Kaplan–Meier curves coupled with a log-rank test, we observed an increased risk of BCR with the high expression of EGFR (75th percentile defined as EGFR^high^; log rank = 5.861, *p* = 0.015) (Figure 2A), while ERBB2 did not show such significance (under 50th percentile defined ERBB2^low^; log rank = 2.441, *p* = 0.118) (Figure 2B). Finally, the low expression of ERBB3 was also an indicator of BCR (25th percentile defined as ERBB3^low^; log rank = 3.768, *p* = 0.052) (Figure 2C).

Univariate Cox regression analyses also demonstrated that EGFR^high^ in continuous (HR = 1.006, CI = 1.003–1.009, *p* = 0.001) or dichotomized (HR = 1.703, CI = 1.0097–2.644, *p* = 0.018) values, as well as dichotomized ERBB3^low^ values (HR = 0.650, CI = 0.418–1.011, *p* = 0.056), showed an association with an increased risk of BCR (less than five years) (Table 2). Continuous ERBB2 and ERBB3 or dichotomized ERBB2 expression values failed to show significance. However, the ERBB family members were not independent of known prognostic clinical parameters (Table 2).

### 3.5. Combining the Expression Levels of EGFR, ERBB2, and ERBB3 Predicts BCR at 5 Years

Since all ERBB family members can dimerize with one another, we evaluated how combinations could be informative of BCR. Therefore, we developed a decision tree model, including dichotomized EGFR, ERBB2, and ERBB3 expression values. Four ERBB status groups were defined (Figure 2D). The decision tree indicated, within a receptor status group, the number of patients with a BCR event. Thereby, we obtained a percentage reflecting the probability to experience the event within the receptor status group. To better represent differences among these newly identified receptor status groups we performed a Kaplan–Meier analysis and observed a significant overall difference (log rank = 25.295, *p* < 0.001) (Figure 2E). Despite significant overall results, some groups are only distinct from one another when compared two-by-two and not from all of the other groups (Figure 2F).

We performed univariate Cox regression analysis using the four ERBB groups and we noted a significant risk of BCR (HR = 1.609, CI = 1.276–2.027, *p* < 0.001) (Table 3). More importantly, patients with PC tissue expressing EGFR^high^, ERBB3^high^, and ERBB2^low^ present a higher 2.189-fold risk of experiencing a BCR that increased to a 5.455-fold higher risk when high ERGF and low ERBB3 expressions were present. In the univariate analyses, the hazard ratio was greater than all clinical parameters. However, in the multivariate analyses, these status groups were not shown to be independent of the clinical parameters (HR = 1.204, CI = 0.930–1.559, *p* = 0.158) (Table 3).

### 3.6. Expression of EGFR, ERBB2, and ERBB3 Can Predict Bone Metastasis Development at 10 Years

Another important endpoint in PC is the development of bone metastasis, which is also recognized as a surrogate for PC mortality. We observed that both EGFR^high^ (log rank = 8.103, *p* = 0.004) and ERBB2^low^ (log rank = 4.539, *p* = 0.033) were significantly associated with an increased risk of developing bone metastasis (Figure 3A,B). However, ERBB3 expression did not confer risk for bone metastasis development (log rank = 0.889, *p* = 0.346) (Figure 3C).

Patients with EGFR^high^ in their PC tissue (continuous or dichotomized values) showed an increased risk of developing bone metastasis when performing Cox regression analyses (Table 4); a risk that reached 3.462-fold (CI = 1.404–1.016, *p* = 0.008) when EGFR expression was dichotomized. In contrast, ERBB2^high^ in PC tissue was associated with a lower risk for bone metastasis (HR = 0.312, CI = 0.101–0.969, *p* = 0.044) and, to a lesser extent, ERBB3^high^ (continuous values) with a protective effect (HR = 0.994, CI = 0.989–1.00, *p* = 0.037) (Table 4).

### 3.7. Combining the Expression Levels of EGFR, ERBB2, and ERBB3 Can Predict Bone Metastasis Development

By incorporating EGFR, ERBB2, and ERBB3 expression in a decision tree model (Figure 3D), four ERBB receptor status groups could be developed to segregate patients based on the risk of developing bone metastasis. Kaplan–Meier analyses performed using these groups revealed an overall log rank of 23.228 with *p* < 0.001 (Figure 3E). The Kaplan–Meier highlights two distinct patient profiles (ERBB2^high^/ERBB2^low^/EGFR^low^) at risk of the development of bone metastases compared to two groups (ERBB2^low^/EGFR^high^ /ERBB3^high^ and ERBB2^low^/EGFR^high^ /ERBB3^low^) (Figure 3F). Moreover, an overall Cox regression analysis, taking into account the combination of all ERBB receptors, showed an increased risk of developing bone metastasis (HR = 2.036, *p* < 0.001) (Table 4). More specifically, our results suggest that patients with tumors expressing ERBB2^low^ and both EGFR^high^ and ERBB3^high^ present a risk that increased by 9.273-fold (CI = 2.311–37.197, *p* = 0.002). The risk of developing bone metastasis reached 14.774 (CI = 2.387–81.237, *p* = 0.002) for patients expressing ERBB2^low^ and ERBB3^low^ coupled with EGFR^high^ and overperformed all clinical parameters.

### 3.8. Expression of EGFR, ERBB2, and ERBB3 Can Predict PC-Specific Mortality

The TF123 TMA series is richly annotated and long-term follow-up allows the evaluation of the ultimate PC endpoint. While EGFR^high^ presented a trend but failing to reach significance (log rank = 2.741, *p* = 0.098) (Figure 4A), both ERBB2^low^ and ERBB3^low^ were significantly associated with PC-specific mortality (log rank = 4.549, *p* = 0.033 and log rank = 4.439, *p* = 0.035, respectively) (Figure 4B,C). Cox regression analysis demonstrated that patients with EGFR^high^ or those with ERBB3^low^ (continuous values) in their PC tissue presented a greater risk of dying from PC (HR = 1.017, CI = 1.010–1.023, *p* < 0.001, and HR = 0.991, CI = 0.985–0.997, *p* = 0.002, respectively). This was also observed for ERBB2^low^ or ERBB3^low^ (dichotomized values) (HR = 0.312, CI = 0.101–0.968, *p* = 0.044 and HR = 0.373, CI = 0.373–0.969, *p* = 0.043, respectively) (Table 5).

### 3.9. Combining the Expression Levels of EGFR, ERBB2, and ERBB3 Can Predict PC Mortality

The decision tree model (Figure 4D) revealed five groups with differential risk of PC-specific mortality as demonstrated by the Kaplan–Meier analyses (log rank = 24.586, *p* < 0.001) (Figure 4E). More precisely, the patients presenting ERBB3^high^/ERBB2^high^ are the group at the least risk of PC-specific mortality, and this group was significantly distinct from all other groups, except those with the ERBB3^high^/ERBB2^low^/EGFR^low^. The group ERBB3^low^/EGFR^high^ was the group with the highest risk of specific PC mortality, then this group was significantly different from all of the other groups, except one (ERBB3^high^/ERBB2^low^/EGFR^high^) (Figure 4F). Cox regression analysis, including the five groups, recapitulated the overall Kaplan–Meier analyses (HR = 1.865, CI = 1.256–2.768, *p* = 0.002) (Table 5). With a hazard ratio greater than all clinical parameters assessed, three groups (orange, grey, and purple) were indicators of patient prognosis. The greatest risk of PC-specific mortality was observed for patients expressing ERBB3^low^ coupled with EGFR^high^ (orange: HR = 36.732, CI = 4.901–275.271, *p* < 0.001), followed by a combination of EGFR^high^/ERBB2^low^/ERBB3^high^ (HR = 11.755, CI = 1.958–70.566, *p* = 0.007), and finally by EGFR^low^/ERBB3^low^ (HR = 5.671, CI = 1.143–28.129, *p* = 0.034). Patients with ERBB2^high^ and ERBB3^high^ were those with a better prognosis. These were not significantly different from patients with EGFR^low^/ERBB2^low^/ERBB3^high^.

## 4. Discussion

In multiple cancers, including PC, a personalized therapeutic approach based on individual tumor characteristics has become an ongoing objective for both treating physicians and patients. Since PC is a heterogeneous disease, and that clinical parameters are insufficient to accurately predict disease outcomes, it is important to identify new tools to aid in clinical decisions and patient management.

In this study, we showed that ERBB family members are associated with a greater risk of BCR. These findings are in line with the literature, where a high expression of EGFR is associated with PC progression [22,32,33], and a low ERBB3 expression, located in the nucleus, is associated with a worse prognosis [27] and an increased risk of BCR. Moreover, this study confirmed the absence or the very low expression of ERBB4 in PC cell lines and primary cancer tissue [34,35]. However, ERBB2 expression in PC progression is more controversial, with one study showing that a high level of ERBB2 is associated with poor prognosis [36], as measured by BCR, while a previous study from our group failed to identify any correlation with BCR [37]. However, in our current study, we identified the importance of ERBB2 as a predictor of eventual bone metastases. These results are in line with those reported in breast cancer studies, where patients presenting with an ERBB2-positive tumor are more likely to metastasize to the bone when compared to the ERBB2-negative group [38]. In addition, several biological studies in pre-clinical prostate models have shown that ERBB2 signaling plays an essential role in the progression from a castration-sensitive to a castration-resistant state associated with bone metastases [39,40]. For example, ERBB2 was shown to play a role in the progression of PC through an increase in angiogenesis, thereby facilitating the dissemination of tumor cells. ERBB2 also confers an androgen independence state, leading to cell survival and proliferation when anti-androgen therapy is used [39]. The contradictory studies looking at ERBB2 expression and the correlation to outcomes in patient tumor tissue could reflect differences in results both in the composition and size of cohorts studied or by differing sources of antibodies used in the studies.

Most studies have highlighted a link between a single member of the ERBB family with PC progression using BCR as an endpoint [22,23,36,37]. When two markers were assessed in the same publication, it was discussed that EGFR is a better predictor of BCR when used alone than in combination [22]. Another study used an heterogenous set of specimens from patients treated (*n* = 29) or not treated with androgen deprivation therapie (*n* = 29) to perform their survival analyses in combining EGFR with ERBB2 [23]. They showed that patients presenting EGFR^high^ and ERBB2^high^ were more likely to experience BCR. In this present study, we combined three out of the four ERBB family members and evaluated their prognostic value against three clinically relevant PC endpoints. We excluded ERBB4 from all analyses in reason of the low expression of this biomarker in prostate cell lines by Western blot and the non-specificity of IF staining for ERBB4 in patient samples. We found that the combination of ERBB3^low^ coupled with EGFR^high^ was associated with the worst prognosis across all endpoints (BCR, development of bone metastasis, and PC-specific survival). We also noted a major role of ERBB2 in the development of bone metastasis and this marker was found to be the first marker used to stratify patients in the decision tree model. The important role of ERBBs in the development of metastases has previously been reported in the literature [40,41,42]. Indeed, in a pre-clinical model using cell line in vivo assays, it was demonstrated that EGFR promotes the survival of PC-circulating tumor cells, while ERBB2 supports cancer cell growth in bones by promoting the RANK signaling pathway [40]. These results support our findings of the association of the ERBB members with worse prognosis.

Since ERBB receptors are differentially expressed in radical prostatectomy specimens, they should form different combinations of dimers (homo-/heterodimer) to activate or inhibit diverse cellular pathways. These different dimerizations, in addition to the fact that the trials did not measure the levels of ERBB receptors, could explain why previous clinical trials studying ERBB family members failed to demonstrate a predictive effect on patient outcomes [43,44,45]. Moreover, it would be interesting to investigate how members of the ERBB family affect anti-androgen therapy. Our results also highlight that the different receptor combinations in the primary tumor are associated with very different outcomes much later in the post treatment setting in these patients. These molecular markers may provide early indicators of patients with worse prognosis requiring more intense follow-up strategies and possibly earlier and more aggressive therapeutic strategies.

## 5. Conclusions

Our results suggest that a different combination of ERBB could be useful to stratify patients following local therapy for PC. We demonstrated that patients presenting with EGFR^high^ coupled with ERBB3^low^ were at a 5-fold increased risk of BCR. Patients expressing ERBB2^low^ had a 14-fold increased risk of developing bone metastasis, and were more than 36 times at higher risk of PC mortality. These biomarkers may become useful in the clinic if they are further validated on larger cohorts.

## Figures and Tables

**Figure 1 cancers-13-01688-f001:**
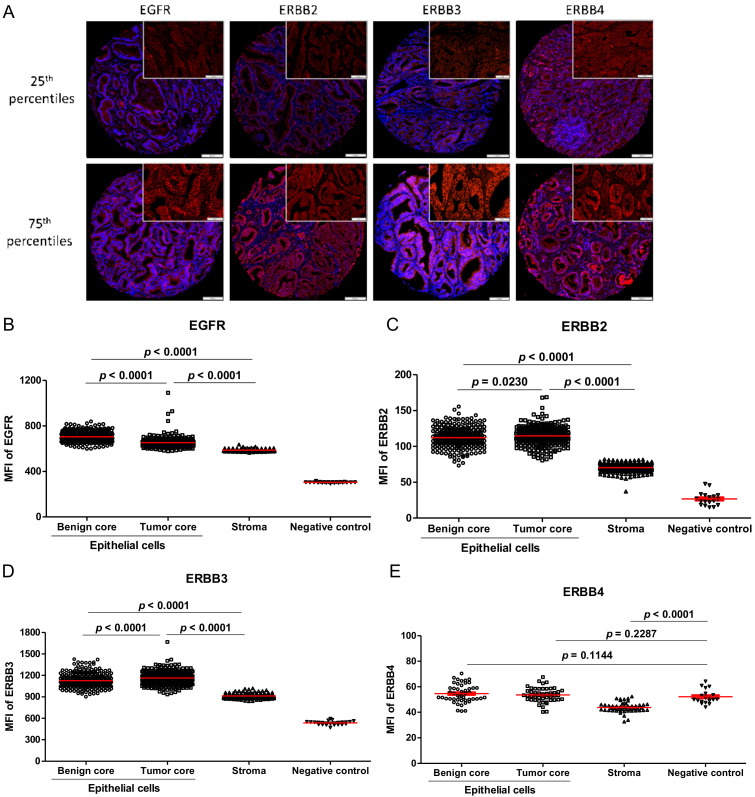
Expression for all ERBB family members in the TF123 TMA series. (**A**) Representative staining of the 25th and 75th percentiles for each protein in epithelial cells of tumor cores. Distribution of (**B**) EGFR, (**C**) ERBB2, (**D**) ERBB3, and (**E**) ERBB4 * expression in the whole PC patient cohort. DAPI (blue) and protein of interest (red). The scale bar at 100 µm is for the whole image and the scale bar at 50 µm for enlarged view. * For ERBB4, only one TMA slide was stained and analyzed.

**Figure 2 cancers-13-01688-f002:**
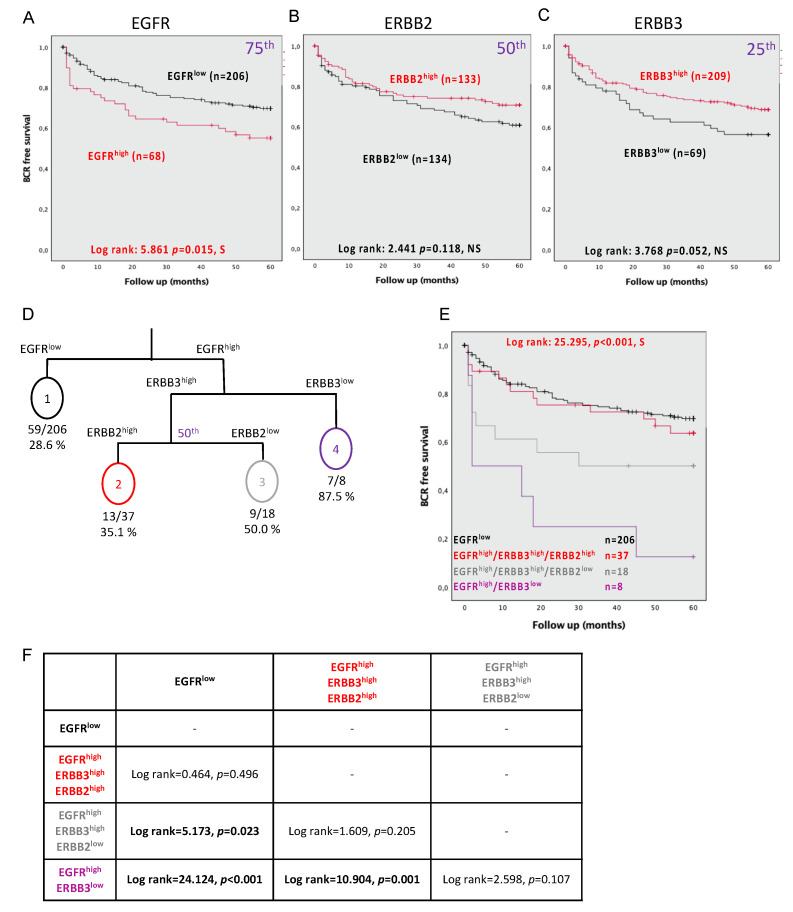
Association of proteins expression in epithelial cells of tumor cores with biochemical recurrence evaluated at five years. Kaplan–Meier curves for (**A**) EGFR with a cutoff at the 75th percentile, (**B**) ERBB2 with a cutoff using the median, and (**C**) ERBB3 with a cutoff at the 25th percentile. (**D**) Survival tree, including EGFR, ERBB2, ERBB3, and BCR in an RPART model. (**E**) Kaplan–Meier plot combines EGFR, ERBB2, and ERBB3 expression. (**F**) Summary table of log rank between each of the conditions of expression of ERBB members. Significance is indicated by log-rank test. *p* < 0.05 was considered significant; NS, not significant; S, significant.

**Figure 3 cancers-13-01688-f003:**
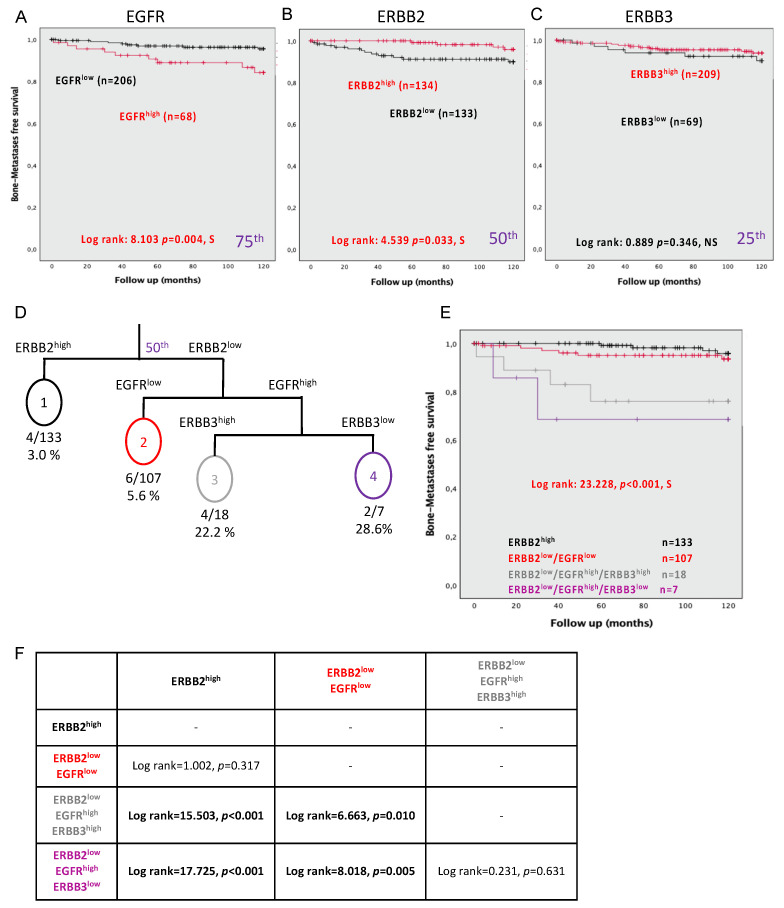
Association of the protein expression in the epithelial cells of tumor cores with the development of bone metastases at 10 years. Kaplan–Meier curves for (**A**) EGFR with a cutoff at the 75th percentile, (**B**) ERBB2 with a cutoff using the median, and (**C**) ERBB3 with a cutoff at the 25th percentile. (**D**) Survival tree, including EGFR, ERBB2, ERBB3, and bone metastases in an RPART model. (**E**) Kaplan–Meier plot combines EGFR, ERBB2, and ERBB3 expression. (**F**) Summary table of log rank between each of the conditions of expression of ERBB members. Significance is indicated by log-rank test. *p* < 0.05 was considered significant; NS, not significant; S, significant.

**Figure 4 cancers-13-01688-f004:**
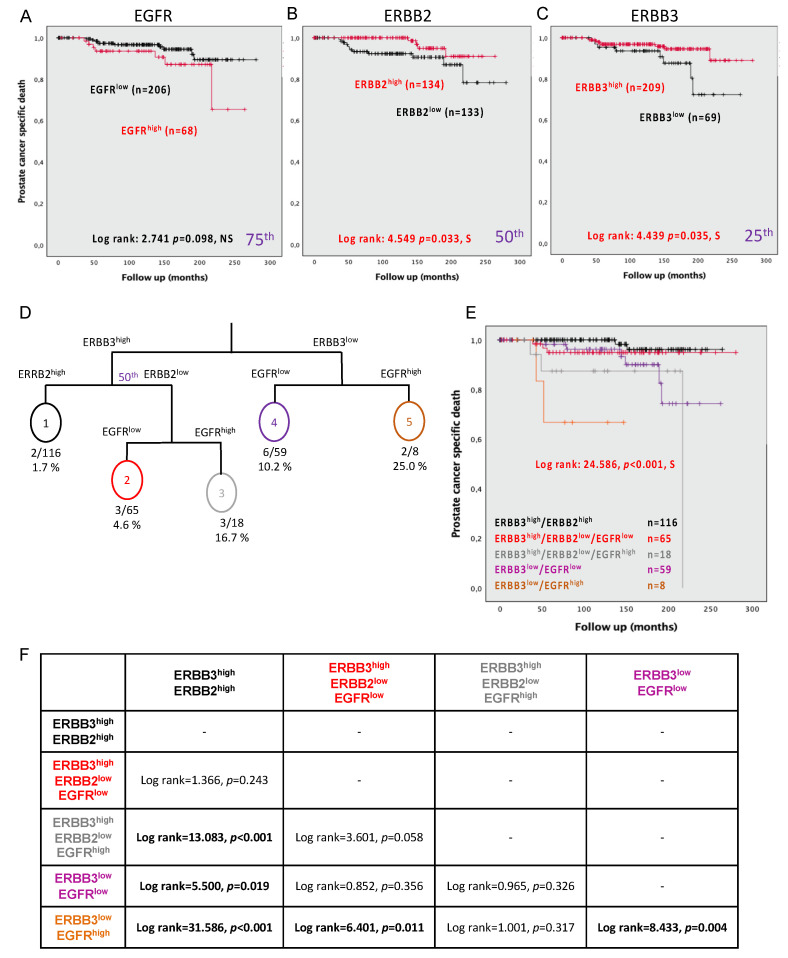
Association of the protein expression in the epithelial cells of tumor cores with prostate cancer-specific death. Kaplan–Meier curves for (**A**) EGFR with a cutoff at the 75th percentile, (**B**) ERBB2 with a cutoff using the median, and (**C**) ERBB3 with a cutoff at the 25th percentile. (**D**) Survival tree, including EGFR, ERBB2, ERBB3, and death by PC in an RPART model. (**E**) Kaplan–Meier plot combines EGFR, ERBB2, and ERBB3 expression. (**F**) Summary table of log rank between each of the conditions of expression of ERBB members. Significance is indicated by log-rank test. *p* < 0.05 was considered significant; NS, not significant; S, significant.

**Table 1 cancers-13-01688-t001:** Clinical and pathological characteristic of the TF123 tissue microarray (TMA) series.

Number of patients	285
Median age at RP, years (IQR)	63 (59–67)
Median PSA at diagnosis, ng/mL (IQR)	7.0 (5.0–10.8)
Pathological TNM	
2	201
3	75
4	9
Gleason score at RP	
≤3 + 3	140
3 + 4	93
4 + 3	19
≥4 + 4	29
Unknown	4
Positive margin	95
Median follow-up, months (IQR)	129 (76–174)
Biochemical recurrence at 5 years	
Number	94
Median time to BCR, months (IQR)	11 (3–26)
Bone Metastasis at 10 years	
Number	18
Median time to bone metastasis, months (IQR)	42 (20–83)
Overall survival	
Alive	236
Death from other cause	31
Death from PC	18
Median time to death specific PC mortality, months (IQR)	68 (49–150)

Abbreviations: TMA, tissue microarray; RP, radical prostatectomy; PSA, prostate-specific antigen; IQR, interquartile range; TNM, tumor, lymph nodes, metastasis; RP, radical prostatectomy; BCR, biochemical recurrence; PC, prostate cancer.

**Table 2 cancers-13-01688-t002:** Univariate and multivariate Cox regression analyses of single receptors using the BCR at five years as an endpoint.

	Univariate			Multivariate with Dichotomized EGFR Expression			Multivariate with Dichotomized ERBB2 Expression			Multivariate with Dichotomized ERBB3 Expression		
	HR [95% CI]		*p*-Value	HR [95% CI]		*p*-Value	HR [95% CI]		*p*-Value	HR [95% CI]		*p*-Value
Age at Dx	0.999	[0.963–1.035]	0.942	-	-	-	-	-	-	-	-	-
PSA at Dx	**1.061**	[1.033–1.089]	**0.001**	1.032	[0.996–1.069]	0.086	1.036	[1.000–1.073]	0.048	1.036	[1.001–1.073]	0.043
Gleason score (4 categories)	**1.852**	[1.549–2.214]	**0.001**	**1.530**	[1.237–1.891]	**0.001**	**1.464**	[1.179–1.819]	**0.001**	**1.517**	[1.231–1.869]	**0.001**
Margin	**3.349**	[2.216–5.062]	**0.001**	**2.617**	[1.655–4.139]	**0.001**	**2.479**	[1.560–3.939]	**0.001**	**2.531**	[1.596–4.015]	**0.001**
pTNM (4 categories)	**2.884**	[2.133–3.900]	**0.001**	**1.555**	[1.047–2.309]	**0.029**	**1.683**	[1.131–2.503]	**0.010**	**1.578**	[1.065–2.340]	**0.023**
EGFR_Tumor_Continuous	**1.006**	[1.003–1.009]	**0.001**	-	-	-	-	-	-	-	-	-
EGFR_Tumor_Dichotomized_75^th^	**1.703**	[1.097–2.644]	**0.018**	1.267	[0.781–2.057]	0.337	-	-	-	-	-	-
ERBB2_Tumor_Continuous	0.987	[0.970–1.004]	0.122	-	-	-	-	-	-	-	-	-
ERBB2_Tumor_Dichotomized_50^th^ *	0.716	[0.468–1.095]	0.123	-	-	-	0.787	[0.503–1.232]	0.295	-	-	-
ERBB3_Tumor_Continuous	0.999	[0.996–1.001]	0.196	-	-	-	-	-	-	-	-	-
ERBB3_Tumor_Dichotomized_25^th^	*0.650*	[0.418–1.011]	*0.056*	-	-	-	-	-	-	0.856	[0.856–1.367]	0.515

Abbreviations: Dx, diagnosis; PSA, prostate-specific antigen; pTNM, pathological tumor, lymph nodes, metastasis; HR, hazard ratio; CI, confidence interval. EGFR_Tumor_Dichotomized designates high expression (over 75%, EGFR^high^) and low (under 75%, EGFR^low^) MFI. * ERBB2_Tumor_Dichotomized designates high expression (over 50% of the median, ERBB2^high^) and low (under 50% of the median, ERBB2^low^) MFI. ERBB3_Tumor_Dichotomized designates high expression (over 25%, ERBB3^high^) and low (under 25%, ERBB3^low^) MFI. *p* < 0.05 is shown in bold, while a *p*-value between ≥0.05 and <0.10 is presented in italics. The symbol ‘-’ indicates that the parameter was not included in the model.

**Table 3 cancers-13-01688-t003:** Univariate and multivariate Cox regression analyses of the ERBB status groups using the BCR at five years as an endpoint.

		Univariate			Multivariate		
		HR [95% CI]		*p*-Value	HR [95% CI]		*p*-Value
Age at Dx		0.999	[0.963–1.035]	0.942	-	-	-
PSA at Dx		**1.061**	[1.033–1.089]	**0.001**	1.029	[0.993–1.067]	0.113
Gleason score		**1.852**	[1.549–2.214]	**0.001**	**1.495**	[1.205–1.856]	**0.001**
Margin		**3.349**	[2.216–5.062]	**0.001**	**2.508**	[1.582–3.976]	**0.001**
pTNM (category)		**2.884**	[2.133–3.900]	**0.001**	**1.559**	[1.044–2.329]	**0.030**
Category	Combined all ERBB members	**1.609**	[1.276–2.027]	**0.001**	1.204	[0.930–1.559]	0.158
	EGFR^low^	**1.000**	-	-	1.000	-	-
	EGFR^high^/ERBB3^high^/ERBB2^high^	1.231	[0.675–2.244]	0.498	1.197	[0.622–2.303]	0.591
	EGFR^high^/ERBB3^high^/ERBB2^low^	**2.189**	[1.085–4.417]	**0.029**	1.122	[0.514–2.449]	0.773
	EGFR^high^/ERBB3^low^	**5.455**	[2.478–12.011]	**0.001**	2.202	[0.915–5.297]	0.078

Abbreviations: Dx, diagnosis; PSA, prostate-specific antigen; cTNM, clinical tumor, lymph nodes, metastasis; pTNM, pathological tumor, lymph nodes, metastasis; HR, hazard ratio; CI, confidence interval. EGFR_Dichotomized designates high expression (over 75%, EGFR^high^) and low (under 75%, EGFR^low^) MFI. ERBB2_Dichotomized designates high expression (over 50% of the median, ERBB2^high^) and low (under 50% of the median, ERBB2^low^) MFI. ERBB3_Dichotomized designates high expression (over 25%, ERBB3 ^high^) and low (under 25%, ERBB3^low^) MFI. Significant results (*p* <0.05) are indicated by bold numbers and results not included are indicated by -.

**Table 4 cancers-13-01688-t004:** Univariate Cox regression analyses of ERBB using the development of bone metastasis at 10 years as an endpoint.

		Univariate		
		HR [95% CI]		*p*-Value
Age at Dx		0.987	[0.907–1.073]	0.757
PSA at Dx		**1.059**	[1.002–1.119]	**0.043**
Gleason score		**3.704**	[2.304–5.955]	**0.001**
Margin		**3.290**	[1.275–8.489]	**0.014**
pTNM (category)		**7.490**	[3.830–14.647]	**0.001**
EGFR_Continuous		**1.012**	[1.007–1.016]	**0.001**
EGFR_Dichotomized		**3.642**	[1.404–9.444]	**0.008**
ERBB2_Continuous		0.976	[0.939–1.016]	0.234
ERBB2_Dichotomized		**0.312**	[0.101–0.969]	**0.044**
ERBB3_Continuous		**0.994**	[0.989–1.000]	**0.037**
ERBB3_Dichotomized		0.622	[0.230–1.683]	0.350
Category	Combined all ERBB members	**2.036**	[1.438–2.881]	**0.001**
	ERBB2^high^	**1.000**	-	**0.001**
	ERBB2^low^/EGFR^low^	1.897	[0.535–6.722]	0.321
	ERBB2^low^/EGFR^high^/ERBB3^high^	**9.273**	[2.311–37.197]	**0.002**
	ERBB2^low^/EGFR^high^/ERBB3^low^	**14.774**	[2.687–81.237]	**0.002**

Abbreviations: Dx, diagnosis; PSA, prostate-specific antigen; cTNM, clinical tumor, lymph nodes, metastasis; pTNM, pathological tumor, lymph nodes, metastasis; HR, hazard ratio; CI, confidence interval. EGFR_Dichotomized designates high expression (over 75%, EGFR^high^) and low (under 75%, EGFR^low^) MFI. ERBB2_Dichotomized designates high expression (over 50% of the median, ERBB2^high^) and low (under 50% of the median, ERBB2^low^) MFI. ERBB3_Dichotomized designates high expression (over 25%, ERBB3^high^) and low (under 25%, ERBB3^low^) MFI. Significant results (*p* < 0.05) are indicated by bold numbers.

**Table 5 cancers-13-01688-t005:** Univariate Cox regression analyses of ERBBs using prostate cancer-specific mortality as an endpoint.

		Univariate		
		HR [95% CI]		*p*-Value
Age at Dx		0.976	[0.896–1.062]	0.569
PSA at Dx		**1.062**	[1.009–1.117]	**0.021**
Gleason score		**3.968**	[2.395–6.573]	**0.001**
Margin		1.731	[0.682–4.391]	0.248
pTNM (category)		**5.655**	[2.928–10.923]	**0.001**
EGFR_Continuous		**1.017**	[1.010–1.023]	**0.001**
EGFR_Dichotomized		2.215	[0.843–5.824]	0.107
ERBB2_Continuous		0.969	[0.933–1.007]	0.107
ERBB2_Dichotomized		**0.312**	[0.101–0.968]	**0.044**
ERBB3_Continuous		**0.991**	[0.985–0.997]	**0.002**
ERBB3_Dichotomized		**0.373**	[0.144–0.969]	**0.043**
Category	Combined all ERBB members	**1.865**	[1.256–2.768]	**0.002**
	ERBB3^high^/ERBB2^high^	**1.000**	-	**0.004**
	ERBB3^high^/ERBB2^low^/EGFR^low^	2.915	[0.487–17.465]	0.241
	ERBB3^high^/ERBB2^low^/EGFR^high^	**11.755**	[1.958–70.566]	**0.007**
	ERBB3^low^/EGFR^low^	**5.671**	[1.143–28.129]	**0.034**
	ERBB3^low^/EGFR^high^	**36.732**	[4.901–275.271]	**0.001**

Abbreviations: Dx, diagnosis; PSA, prostate-specific antigen; cTNM, clinical tumor, lymph nodes, metastasis; pTNM, pathological tumor, lymph nodes, metastasis; HR, hazard ratio; CI, confidence interval. EGFR_Dichotomized designates high expression (over 75%, EGFR^high^) and low (under 75%, EGFR^low^) MFI. ERBB2_Dichotomized designates high expression (over 50% of the median, ERBB2^high^) and low (under 50% of the median, ERBB2^low^) MFI. ERBB3_Dichotomized designates high expression (over 25%, ERBB3^high^) and low (under 25%, ERBB3^low^) MFI. Significant results (*p* < 0.05) are indicated by bold numbers.

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
