# Peer review of "Expression of ERBB Family Members as Predictive Markers of Prostate Cancer Progression and Mortality"

_cancers, 2021, doi:10.3390/cancers13071688_

Round 1
Reviewer 1 Report
Expression of ERBB family members as predictive markers of 2 prostate cancer progression and mortality
Overview
The authors studied expression of all members of the ERBB family in 27 PC tissue from a large cohort and determined their contribution, alone or in combination, as 28 prognostic markers and observed that ERBB family members are associated with an increased risk of PC 35 progression and mortality.
Novelty
Several studies have shown the importance of ERBB in prostate cancer prognosis as well as its possibility as a biomarker (Brizzolara et al., 2017, Morote et al., 1999, Ratan et al., 2003, Koumakpayi et al., 2010).
Major comments
- What is the mechanistic advance in this study? EGFR and ERBB have been extensively studied for their importance in PC.
- Is there an importance of this observation in castration resistant prostate cancer? Did the authors study patients with castration resistant cancer?
- “EGFRhigh coupled with ERBB3low were at 5-fold increased risk of BCR. Patients expressing ERBB2low had a 14-fold increased risk of developing bone metastasis, and were more than 36 times at higher risk of PC mortality” What is the molecular mechanism behind this observation? How did these specific biomarker- based tumors respond to therapy? Is BCR not associated with PC mortality and metastasis in the study, the observation could use more explanation.
- How do these biomarkers compare to existing biomarkers for PC?
- Were there drugs or other therapy that these biomarker based tumors were found to be particularly sensitive or resistant to?
- How was the protein expression of the TMA findings confirmed other than IHC? Were the primary samples studied by RT-PCR or western blot?
Reviewer 2 Report
In the manuscript (cancers-1142625), the authors assessed the role of ERBB family members as predictive markers of prostate cancer progression and mortality. The author did characterize the expression of the ERBB family in prostate cancer cell lines and also in human patient samples.
- The manuscript is very well written and addresses the objective precisely.
- The author observed that all antibodies showed a single band (Figure S1-A). EGFR protein levels were higher in DU145, 22Rv1, and 213 PC3 cells when compared to the LNCaP cell line, which I infer that ERBB somewhere modulates the expression of AR, compared to ERBB2 and ERBB3. My only concern to know why Jurkat T lymphoma cells were used in the experiment rather than using RWPE-1.
- It is interesting to know that overall Cox-regression analysis of all ERBB receptors showed an increased risk of developing bone metastasis. I am wondering if the author here talks about all means along with ERBB4, which did not show any or less expression in patient cohorts.
Reviewer 3 Report
The authors evaluated the expression of ERBB family members in PC tissue from a cohort of patients with a long term follow-up, in order to determine their contribution as prognostic markers. The study design and the presentation of the results are remarkable. I think that these findings could impact on future tailored tratments for PC.
Author Response
We would like to thank the reviewer for this comment and appreciation of our manuscript.
Reviewer 4 Report
In this manuscript titled "Expression of ERBB family members as predictive markers of prostate cancer progression and mortality" the authors investigate the different combinations of ERBB that could be useful to stratify patients following local therapy for PC. The manuscript is meticulously prepared and results well demonstrated. The authors show that patients presenting with an EGFR high coupled with ERBB3low were at a 5-fold increased risk of BCR. Patients expressing ERBB2low had a 14-fold increased risk of developing bone metastasis and were more than 36 times at higher risk of PC mortality. Indeed these biomarkers could be of use in the clinic.
The authors base the results with samples evaluated from various patient cohorts which is well justified.
The caveat is that the authors do not show in in-vitro data to complement their clinical study such as knockdown experiments to drive the point that if EGFR is eliminated and reversal of ERBB2 and ERBB3 could attenuate cancer progression, or confirm if their presence has a different effect.
The authors have not discussed the role of EGFR, ERBB2, and ERBB3 from an Androgen receptor scenario when most prostate cancers have AR initially which disappears and reappears. Does expression of AR or AR activation contribute to the role played by EGFR, ERBB2, and ERBB3?
The authors will have to discuss how these high EGFR, Low ERBBs affect anti-androgen therapy.
The Clinical samples evaluated in the TF123 TMA series are composed of 285 PC patients the authors need to show AR expression.
The Western Blotting data in the supplemental figure should include AR and better loading control.
Round 2
Reviewer 4 Report
The authors have provided a convincing explanation and the manuscript may be accepted for publication.